# A Hierarchical Control Scheme for Adaptive Cruise Control System Based on Model Predictive Control

**Hongyuan Mu** [1], **Liang Li** [1,*], **Mingming Mei** [1] **and Yongtao Zhao** [2]

[1] State Key Laboratory of Automotive Safety and Energy, Tsinghua University, Beijing 100084, China; mhy17@mails.tsinghua.edu.cn (H.M.); mmm19@mails.tsinghua.edu.cn (M.M.)
[2] School of Engineering and Technology, China University of Geosciences, Beijing 100083, China; zytzyy@email.cugb.edu.cn
[*] Correspondence: liangl@tsinghua.edu.cn

**Abstract:** An adaptive cruise control (ACC) system can improve safety and comfort during driving by taking over longitudinal control of the vehicle. It requires the coordination between the upper-layer controller and the lower-layer actuators. In this paper, a hierarchical anti-disturbance cruise control architecture based on electronic stability control (ESC) system is proposed. The upper-layer controller outputs the desired longitudinal acceleration or deceleration to the lower-layer actuators. In order to improve the accuracy of model prediction and achieve the coordinated control of multiple objectives, an upper-layer model prediction cruise controller is established based on feedback control and disturbance compensation. In addition, based on the hydraulic control unit (HCU) model and the vehicle longitudinal dynamics model, a lower-layer nonlinear model predictive deceleration controller is proposed in order to solve the problems of pressure fluctuations and the low accuracy of small decelerations when ESC is used as the actuator for the ACC system. Finally, the simulation and experimental tests were carried out. The results show that the proposed control architecture can improve the stability and comfort of the cruise control process. Moreover, compared with the traditional PID deceleration controller, it effectively improves the deceleration control accuracy.

**Keywords:** adaptive cruise control; model predictive control; brake-by-wire; deceleration control

## 1. Introduction

With the continuous progress of automotive technology, people have increasingly high requirements for automotive safety and comfort. The adaptive cruise control (ACC) system is one of the most advanced driving assist systems (ADAS) that helps the driver by taking over the accelerator and brake pedals, automatically controlling the relative speed and distance between the ego vehicle and the front vehicle [1,2]. Not only does it effectively improve the safety and comfort of the driving process but also traffic efficiency [2–6]. ACC systems mostly adopt a hierarchical control algorithm. The upper-layer controller senses the traffic situation ahead through sensors, such as radar and cameras; judges the required longitudinal acceleration; and sends the command to the lower-layer drive-by-wire (DBW) system and brake-by-wire (BBW) system for longitudinal dynamics control [7].

The upper-layer controller for ACC systems have been studied since the 1960s, and researchers have designed various cruise controllers. Zhang and Pradhan et al. [8,9] designed the traditional PID controller to adjust the relative speed and relative distance during cruise. Chaturvedi et al. [10] designed an optimized PID controller based on particle swarm optimization technology and teaching–learning optimization technology, which reduces the overshoot rate and the rise time of the system. Sawant et al. [11] proposed a sliding mode controller based on the disturbance observer to control the ACC system to overcome the model disturbance and analyzed the stability of the system. Li et al. [12] designed a terminal sliding mode controller, which improves the robustness of the cruise controller to modeling uncertainties and external disturbances and improves the convergence rate of the

system. In recent years, in order to achieve multi-objective coordinated control, the model predictive control (MPC) has been applied to cruise controllers. Ma and Nie et al. [13,14] established an ACC controller based on MPC, which considered various constraints, such as economy, safety, and comfortability characteristics, and was proven to improve economic efficiency through simulation. Xu et al. [15] proposed a novel ACC strategy based on a hierarchical framework, established a sliding acceleration identification model to improve the robustness of the upper-layer MPC controller and built an iterative learning lower-layer controller to reduce vehicle speed fluctuations.

Most of the BBW systems currently used in passenger cars are electronic stability control (ESC) systems, which are often used as the actuators for ADAS, such as the ACC system. The hydraulic control unit (HCU) of ESC is mainly composed of solenoid valves, coil, plunger pumps, and DC motors [16,17]. The most common method is to use pulse width modulation (PWM) to control the motor and the coil to drive the plunger pump and solenoid valve, respectively. By adjusting the hydraulic pressure of the wheel cylinder, the longitudinal deceleration of the vehicle can be controlled. However, HCU has highly nonlinear characteristics, and there are problems such as pressure overshoot and noise.

Early wheel cylinder pressure (WCP) controllers generally used the PID method to control the duty cycle of the solenoid valve [18], but there were problems in pressure overshoot and rapidity. Certain improved PID control methods have recently been developed. In recent years, with the research on the model of HCU, researchers have proposed a series of novel controllers. Meng and Fang et al. [19,20] established the mathematical model of the key components of HCU and analyzed the impact of component parameters on brake pressure control using simulation and experimental methods. Han et al. [21] proposed a sliding mode pressure controller based on a pressure estimator to track the desired pressure without pressure sensors. Zhao and Braun et al. [22,23] realized the observation of solenoid valve spool position and WCP control based on the sliding mode observer and controller. Zhao et al. [24] established a pressure controller for commercial vehicles based on the MPC theory and carried out a simulation and experimental verification.

This paper takes passenger cars as the research object and designs a hierarchical cruise controller scheme. Considering the interference of vehicle parameters, sensor measurement errors, road disturbances, and other factors during driving, combined with the principles of feedback control and disturbance compensation, a robust upper-layer MPC controller is proposed for coordinating multiple control objectives. In addition, according to the physical model of HCU and the vehicle dynamics, a lower-layer deceleration controller is proposed based on nonlinear model predictive control (NMPC), which controls the duty of the motor and the coil, respectively, and realizes the precise hydraulic pressure adjustment of the wheel cylinder.

The structure of this paper is organized as follows. In Section 2, the cruise system model, vehicle dynamics model, and BBW system model are presented. In Section 3, the hierarchical control architecture of the ACC system is designed based on the MPC theory. In Section 4, the simulation and experiment are carried out to verify the effectiveness of the proposed method. Finally, the conclusions are given in Section 5.

## 2. System Modeling

### 2.1. Adaptive Cruise Control System Modeling

During the cruise process, the kinematic relationship between the ego vehicle and the front vehicle is shown in Figure 1. $v_{ego}$ and $a_{ego}$ represent the velocity and acceleration of the ego vehicle, respectively. $v_o$ and $a_o$ represent the velocity and acceleration of the front vehicle, respectively. $x_{rel}$ is the relative distance between the two vehicles.

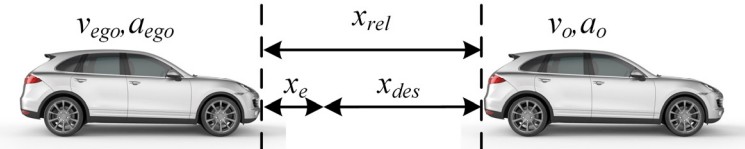

**Figure 1.** Schematic diagram of vehicle cruising process.

The kinematic relationship between the ego vehicle and the front vehicle can be described as follows:

$$\begin{cases} x_e = x_{des} - x_{rel} \\ v_e = v_o - v_{ego} \end{cases} \tag{1}$$

where $x_{des}$ represents the desired distance between two vehicles. $x_e$ and $v_e$ represent the distance error and the velocity error, respectively. The desired velocity of the ego vehicle is $v_o$.

During the cruising process, based on the constant time headway $T_h$, the desired distance $x_{des}$ can be calculated as follows [25]:

$$x_{des} = v_{ego} T_h + d_{safe} \tag{2}$$

where $d_{safe}$ is the safe distance when the ego vehicle stops.

Considering the delay characteristics of the vehicle drive system [26], the relationship between the actual longitudinal acceleration $a_{ego}$ and the desired longitudinal acceleration $a_{des}$ is approximately considered to be a first-order system, and its discretization expression is described as follows:

$$a_{ego}(k+1) = \left( \frac{\tau}{\tau + T_s} \right) a_{ego}(k) + \left( \frac{T_s}{\tau + T_s} \right) a_{des}(k) \tag{3}$$

where $\tau$ is the time delay coefficient, $T_s$ is the sampling time.

Selecting $x(k) = [x_e(k) \ v_{rel}(k) \ a_{ego}(k)]^{\mathrm{T}}$ as the state vector of the longitudinal vehicle model and $u = a_{des}(k)$ as the input vector, the discrete-time state space equation of ACC system can be established as follows:

$$\begin{cases} x(k+1) = Ax(k) + Bu(k) + \Gamma d(k) \\ y(k) = Cx(k) \\ d(k) = a_0(k) \end{cases} \tag{4}$$

where $A = \begin{bmatrix} 1 & -T_s & T_h T_s \\ 0 & 1 & -T_s \\ 0 & 0 & 1 - T_s/\tau \end{bmatrix} B = \begin{bmatrix} 0 \\ 0 \\ T_s/\tau \end{bmatrix} C = \begin{bmatrix} 1 & 0 & 0 \\ 0 & 1 & 0 \end{bmatrix} \Gamma = \begin{bmatrix} 0 \\ T_s \\ 0 \end{bmatrix}.$

### 2.2. Vehicle Longitudinal Dynamics Modeling

Neglecting the influence of the tire slip and considering that the slope angle is very small when the vehicle is running normally, the equation for the longitudinal dynamics of a vehicle can be obtained as follows [27]:

$$\delta m a_{ego} = \frac{T_{tq} i_g i_0 \eta_T}{r} - \frac{P_w K_P}{r} - Gf - Gi - \frac{1}{2} C_D A \rho_a v_{ego}^2 \tag{5}$$

where $\delta$ represents the rotational mass coefficient; $m$ represents the mass of vehicle; $T_{tq}$ represents the engine torque; $i_g$ and $i_0$ are the gear ratios of the transmission and main retarder, respectively; $\eta_T$ represents the efficiency of the transmission system; $r$ is the radius of the wheel; $G$ is the gravity of the vehicle; $f$ is the rolling resistance coefficient; $i$ is the slope of the road; $C_D$ is the air resistance coefficient; $A$ is the windward area of the vehicle; $\rho_a$ is the air density; $P_w$ is the WCP; and $K_P$ is a gain coefficient.

### 2.3. Hydraulic Control Unit Modeling

2.3.1. Wheel Cylinder and Braking Pipeline Model

Passenger vehicles generally adopt a hydraulic disc braking system, and the HCU is the main part that realizes WCP control. The two hydraulic circuits in the HCU have the same structure and are connected to the two oil outlets of the master cylinder, respectively. Taking one hydraulic circuit of the HCU as an example, its structural diagram is shown in Figure 2, in which the isolate valve (IV) and the apply valve (AV) are normally open, and the prime valve (PV) and release valve (RV) are normally closed. During the pressurization process, the PV is energized to open, the IV is energized to close, and the motor is energized to drive the plunger pump to pump the brake fluid from the low-pressure area into the high-pressure area. During the decompression process, only the IV is energized to control the brake fluid from the high-pressure area to the low-pressure area with a certain duty. During the pressure maintenance process, only the IV is energized to close and maintain the pressure of the high-pressure area.

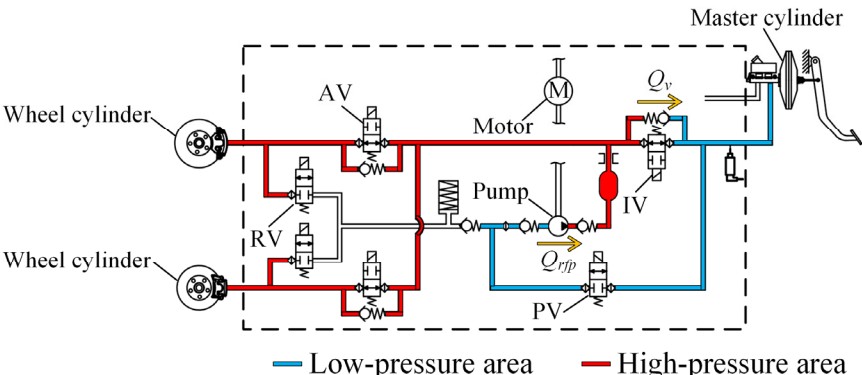

**Figure 2.** Schematic diagram one hydraulic circuit of HCU.

Neglecting the compressibility of brake fluid, the relationship between WCP and the flow rate of the brake fluid can be described as follows:

$$\dot{P}_w(t) = K\left(Q_{rfp} - Q_v\right) \tag{6}$$

where $K$ is the volumetric stiffness of the wheel cylinder; $Q_{rfp}$ and $Q_v$ are the volumetric flow rate flowing in from the plunger pump and the volumetric flow rate flowing out from the IV, respectively.

2.3.2. Motor and Plunger Pump Model

The schematic diagram of the positional relationship between the motor and the plunger pump is shown in Figure 3. The motor drives the plunger pumps symmetrically distributed on both sides via the eccentric camshaft. Point $O$ is the rotation center of the camshaft whose eccentricity is $d_0$. $d$ is the inner diameter of the plunger, and $\theta$ is the rotation angle of the motor shaft. Within a working cycle, the plunger reciprocates once to achieve the suction and pumping of brake fluid.

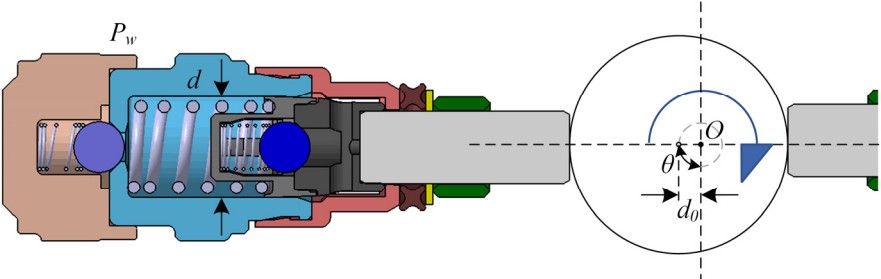

**Figure 3.** Schematic diagram of motor and plunger pump.

The motor is a permanent magnet DC motor, and its speed characteristics can be described as follows:

$$n = \frac{U_a}{C_e \Phi} - \frac{R_a}{C_e C_T \Phi^2}(T_L + T_0) \tag{7}$$

where $n$ is the speed of the motor; $U_a$ is the input voltage of the motor; $C_e$ and $C_T$ are the electromotive force constant and torque constant, respectively; $R_a$ is the armature circuit resistance; $\Phi$ is the magnetic flux; $T_L$ is the load torque; and $T_0$ is the no-load torque.

According to the positional relationship between the plunger pump and the motor, and ignoring the torque fluctuation in each cycle, the load torque can be calculated as follows:

$$\overline{T}_L = P_w d^2 d_0 \tag{8}$$

where $P_w$ is the outlet pressure of the plunger pump, which is equal to the WCP.

Considering the leakage of the plunger pump and ignoring the flow fluctuation in each cycle, the corrected average flow rate of the plunger pump is calculated as follows:

$$\begin{aligned} Q_{rfp} &= \frac{\pi}{120} d^2 d_0 n \eta_v \\ &= \frac{\pi}{120} d^2 d_0 \eta_v \left[ \frac{U_a}{C_e \Phi} - \frac{P_w d^2 d_0 R_a}{C_e C_T \Phi^2} - \frac{R_a T_0}{C_e C_T \Phi^2} \right] \end{aligned} \tag{9}$$

where $\eta_v$ represents the efficiency of the plunger pump.

According to (7)–(9), the average flow rate of a single plunger pump can be described as follows:

$$Q_{rfp} = f_p(U_a, P_w) \tag{10}$$

### 2.3.3. Solenoid Valve and Coil Model

The structure of the IV is shown in Figure 4, and the coil is set outside the magnetic isolation tube of the IV. The spool of the IV is subjected to multiple forces, such as the electromagnetic force, spring force, and hydraulic force, etc. [28]. Only the electromagnetic force can be controlled by the current of the coil.

The current response time of the electromagnetic coil and the response time of the spool movement are obviously shorter than the coil voltage control cycle. Therefore, ignoring the coil inductance and the counter electromotive force generated by the movement of the spool, the relationship between the coil current $i$ and the input voltage $U_c$ is shown as follows:

$$U_c = i \cdot R \tag{11}$$

where $R$ is the coil resistance.

Neglecting the viscous force and friction force during the movement of the spool, the kinetic equation of the spool can be obtained as follows:

$$\ddot{z} = \frac{1}{m_s}(F_m - F_s - F_k) \tag{12}$$

where $z$ is the displacement of the spool, $z = 0$ mm when the spool is in contact with the valve seat, and the IV is fully closed. $m_s$ is the mass of the spool, and $F_m$, $F_s$, and $F_k$ represent the electromagnetic force, hydraulic force, and spring force on the spool, respectively.

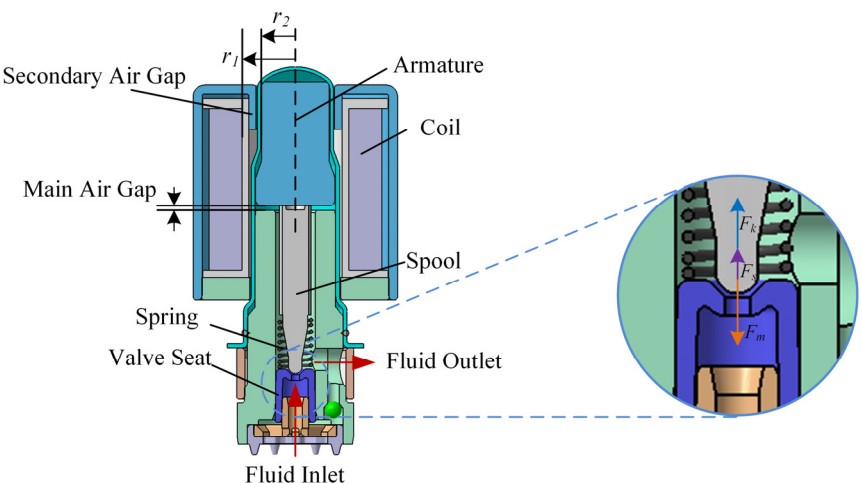

**Figure 4.** Schematic diagram of the isolate valve.

According to Maxwell's equations, the main air gap reluctance and secondary air gap reluctance are mainly considered, and the electromagnetic force can be calculated as follows [28]:

$$F_m = \frac{\psi(i,z)^2}{2\mu_0 S_0} = \frac{i^2 \cdot N^2}{2\mu_0 S_0 \left( \frac{z+z_0}{\pi\mu_0 S_0} + \frac{\ln(r_1/r_2)}{2\pi\mu_0 l_n} \right)} \quad (13)$$

where $\psi(i,z)$ represents the flux linkage, $\mu_0$ is the vacuum permeability, $S_0$ is the area of air gap; $N$ is the number of turns of coil; $z_0$ is the length of the main air gap when the IV is fully closed. $r_1$ and $r_2$ represent the radius of the armature and the secondary air gap, respectively; $l_n$ represents the width of the secondary air gap.

In order to calculate the hydraulic force on the spool, the control volume is selected as shown in Figure 5. Applying Reynolds transport theorem, the hydraulic force can be calculated as follows [29]:

$$F_s = \rho_v Q_v v_0 - \rho_v Q_v v_1 \cos\alpha + P_0 A_0 + cP_1 A_t \sin\alpha - P_1 A_1 \cos\alpha \quad (14)$$

where $A_0$, $A_1$, and $A_t$ are the area of the valve inlet section, the valve throttle, and valve seat, respectively, and $A_2$ is the area of the valve outlet; and $P_0$, $P_1$, and $P_2$ are the pressures at $A_0$, $A_1$, and $A_2$, respectively. Assume that under the pressurization condition, $P_0$ is equal to $P_W$ and $P_2$ is 0. $v_0$ and $v_1$ are the average velocity of the fluid at $A_0$ and $A_1$, respectively. $\rho_v$ is the density of the brake fluid; $Q_v$ is the flow rate flowing through the IV; $\alpha$ is the half-cone angle of the valve seat; and $c$ is the correction coefficient for uneven pressure distribution at $A_t$.

Based on the geometric structure, each area can be calculated as follows:

$$\begin{cases} A_0 = \frac{\pi}{4} d_v^2 \\ A_1 = \pi \left( z^2 \cdot \sin^2\alpha \cdot \cos\alpha + 2R_s z \cdot \sin\alpha \cdot \cos\alpha \right) \\ A_t = \pi \left( (R_s + z\sin\alpha)^2 \cdot \frac{\cos^2\alpha}{\sin\alpha} - \frac{d_v^2}{4\sin\alpha} \right) \end{cases} \quad (15)$$

where $R_s$ is the radius of the ball at the top of the spool, and $d_v$ is the inlet diameter of the valve seat.

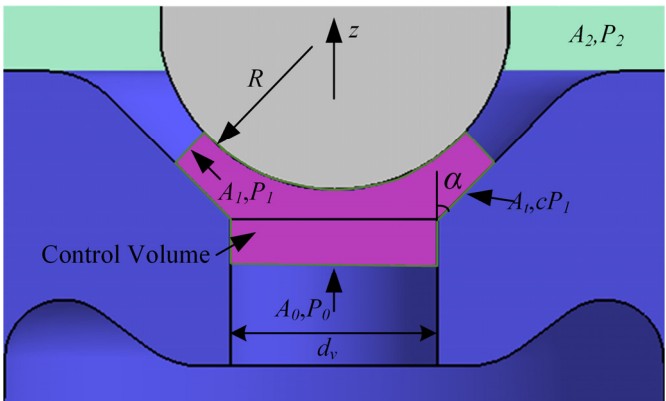

**Figure 5.** Schematic diagram of the control volume of the isolate valve.

Define $\Delta P = P_0 - P_2$ as the pressure drop across the valve. Calculate the valve flow rate according to the orifice throttling formula as follows:

$$Q_v = c_d A_1 \sqrt{2\Delta P / \rho_v} \tag{16}$$

where $c_d$ is the flow coefficient of the valve.

According to the Bernoulli equation and continuity principle, we can obtain:

$$Q_v = A_0 v_0 = A_1 v_1 = A_2 v_2 \tag{17}$$

$$P_1 = P_2 + c_d{}^2 \Delta P \left[ (1 + \zeta) \frac{A_1{}^2}{A_2{}^2} - 1 \right] \tag{18}$$

where $\zeta$ is the energy dissipation coefficient of the valve.

Considering that $A_1$ is much smaller than $A_2$, the pressure at the valve throttle can be calculated as follows:

$$P_1 = P_2 - c_d{}^2 \Delta P \tag{19}$$

Substituting (16)–(19) into (14), the steady-state hydrodynamic forces can be obtained as follows:

$$F_s = P_0 \left[ \frac{2c_d^2 A_1^2}{A_0} - c_d^2 A_1 \cos \alpha - c c_d^2 A_t \sin \alpha + A_0 \right] \tag{20}$$

The spring force on the spool can be calculated as follows:

$$F_k = F_0 - K_k z \tag{21}$$

where $K_k$ is the spring stiffness, $F_0$ is the spring force when the IV is fully closed.

As mentioned above, the response time of the spool movement is much shorter than the coil voltage control cycle. Therefore, assuming that the spool acceleration $\ddot{z} = 0$, substituting (13), (20) and (21) into (12), the spool displacement $z(U_c, P_w)$ can be obtained. Then, according to (16), the flow rate $Q_v$ can be obtained as follows:

$$Q_v = c_d A_1(z(U_c, P_w)) \sqrt{\frac{2P_w}{\rho}} = f_v(U_c, P_w) \tag{22}$$

## 3. Controller Design

The ACC controller adopts a hierarchical control structure, as shown in Figure 6, including an upper-layer controller and a lower-layer controller. The upper-layer controller calculates the desired acceleration $a_{des}$ of the vehicle according to the current motion state of the vehicle and the relative motion state with the front vehicle and transmits it to the

lower-layer controller. According to the instructions of the upper-layer controller, the lower-layer controller controls the engine torque through the interface of the engine management system (EMS) and adjusts the WCP by controlling the HCU to realize the tracking of the desired acceleration $a_{des}$.

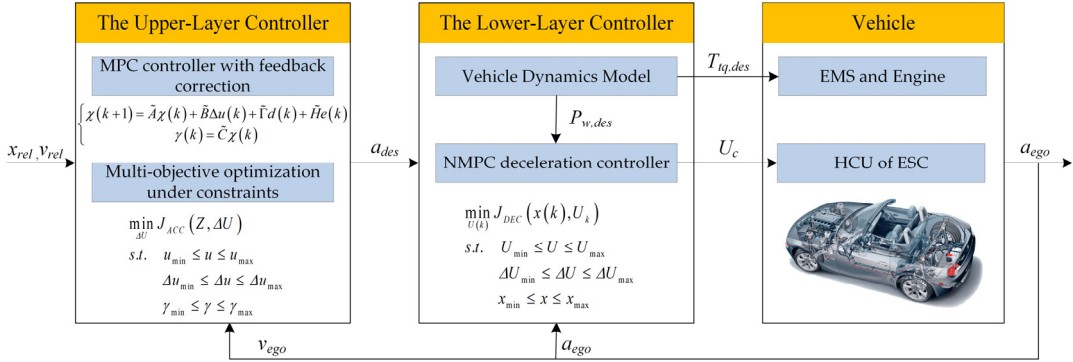

**Figure 6.** The overall architecture of the hierarchical ACC controller.

### 3.1. The Design of the Upper-Layer Controller

In the car-following process, changes in a vehicle's mass, road surface, and other conditions will have a certain impact on the accuracy of the model prediction, and the measurement accuracy of the sensor will also lead to a certain error. In order to improve the robustness of system, the feedback correction method is applied to the MPC controller.

According to the system state measurement value $x(k)$ at time $k$ and the one-step system state prediction value $x(k|k-1)$ at time $k-1$, construct the system state prediction error as follows:

$$e(k) = x(k) - x(k|k-1) \tag{23}$$

Define a new system state variable $\chi(k) = [x(k)\ u(k-1)]^{\mathrm{T}}$, the augmented state-space equation with feedback correction term can be obtained as follows:

$$\begin{cases} \chi(k+1) = \widetilde{A}\chi(k) + \widetilde{B}\Delta u(k) + \widetilde{\Gamma}d(k) + \widetilde{H}e(k) \\ \gamma(k) = \widetilde{C}\chi(k) \end{cases} \tag{24}$$

where $\widetilde{A} = \begin{bmatrix} A & B \\ 0 & I \end{bmatrix}, \widetilde{B} = \begin{bmatrix} B \\ I \end{bmatrix}, \widetilde{C} = \begin{bmatrix} C & 0 \end{bmatrix}, \widetilde{\Gamma} = \begin{bmatrix} \Gamma \\ 0 \end{bmatrix}, \widetilde{H} = \begin{bmatrix} H \\ 0 \end{bmatrix}, \Delta u(k) = u(k) - u(k-1)$, $H$ is the feedback correction coefficient matrix, and $I$ is the identity matrix.

Define the prediction time domain of the MPC controller as $p$, and the control time domain as $m$, the system prediction model with feedback correction items can be obtained as follows:

$$Z = S_\chi \chi(k) + S_{\Delta U} \Delta U + S_d d(k) + S_e e(k) \tag{25}$$

$$\text{where} \quad Z = \begin{bmatrix} \gamma(k+1) \\ \vdots \\ \gamma(k+m) \\ \vdots \\ \gamma(k+p) \end{bmatrix}, S_\chi = \begin{bmatrix} \widetilde{C}\widetilde{A} \\ \vdots \\ \widetilde{C}\widetilde{A}^m \\ \vdots \\ \widetilde{C}\widetilde{A}^p \end{bmatrix}, S_d = \begin{bmatrix} \widetilde{C}\widetilde{\Gamma} \\ \vdots \\ \sum_{i=0}^{m-1} \widetilde{C}\widetilde{A}^i\widetilde{\Gamma} \\ \vdots \\ \sum_{i=0}^{p-1} \widetilde{C}\widetilde{A}^i\widetilde{\Gamma} \end{bmatrix}, S_e = \begin{bmatrix} \widetilde{C}\widetilde{H} \\ \vdots \\ \widetilde{C}\widetilde{A}^{m-1}\widetilde{H} \\ \vdots \\ \widetilde{C}\widetilde{A}^{p-1}\widetilde{H} \end{bmatrix},$$

$$S_{\Delta U} = \begin{bmatrix} \widetilde{C}\widetilde{B} & 0 & \cdots & 0 \\ \vdots & \vdots & \vdots & \vdots \\ \widetilde{C}\widetilde{A}^{m-1}\widetilde{B} & \widetilde{C}\widetilde{A}^{m-2}\widetilde{B} & \cdots & \widetilde{C}\widetilde{B} \\ \vdots & \vdots & \vdots & \vdots \\ \widetilde{C}\widetilde{A}^{p-1}\widetilde{B} & \widetilde{C}\widetilde{A}^{p-2}\widetilde{B} & \cdots & \widetilde{C}\widetilde{A}^{p-m}\widetilde{B} \end{bmatrix}, \Delta U = \begin{bmatrix} \Delta u(k) \\ \vdots \\ \Delta u(k+m-1) \\ \vdots \\ \Delta u(k+p-1) \end{bmatrix}.$$

Considering the tracking capability and comfort of the ACC system comprehensively, the multi-objective cost function $J_{ACC}$ is designed as follows [30]:

$$J_{ACC}(Z, \Delta U) = \left(Z - Z_{ref}\right)^T Q\left(Z - Z_{ref}\right) + \Delta U^T R \Delta U \tag{26}$$

where $Z_{ref}$ is the reference value of the system state, and $Q$ and $R$ are the weight coefficient matrices of the predicted output and system control input, respectively.

The vehicle's BBW system and engine system limit the vehicle's dynamic response characteristics, such as maximum speed, maximum acceleration, etc. In addition, the safety objective of the ACC system, as well as the riding comfort of the driver and passengers should also be considered. In summary, the optimization problem of the upper controller of the ACC system under the constraints can be described as follows:

$$\begin{aligned} &\min_{\Delta U} J_{ACC}(Z, \Delta U) \\ &s.t. \quad u_{\min} \leq u \leq u_{\max} \\ &\qquad \Delta u_{\min} \leq \Delta u \leq \Delta u_{\max} \\ &\qquad \gamma_{\min} \leq \gamma \leq \gamma_{\max} \end{aligned} \tag{27}$$

where $u_{\min}$ and $u_{\max}$ and $\Delta u_{\min}$ and $\Delta u_{\max}$ are the upper and lower limits of the expected acceleration and the control output change rate, respectively. $\gamma_{\min}$ and $\gamma_{\max}$ are the upper and lower limits of each state variable, such as the distance error and the velocity error.

### 3.2. The Design of the Lower-Layer Controller

As the lower-layer actuators system, the DBW system and BBW system are used to accurately execute the desired longitudinal acceleration calculated by the upper-layer controller. However, changes in external conditions, such as changes in road conditions and changes in air resistance caused by vehicle speed, make it difficult for the actuator system to track the required deceleration accurately. In addition, the BBW system using ESC has strong nonlinear characteristics, which also increases the difficulty of control. In order to achieve precise control of vehicle acceleration and deceleration, a nonlinear model predictive controller (NMPC) is designed in this section to control the engine torque and the duty of the IV and the motor in coordination.

When the desired longitudinal acceleration is positive, the WCP is zero, and the desired engine torque $T_{tq,des}$ can be calculated according to (5) as follows:

$$T_{tq,des} = \frac{\delta m a_{des} + Gf + Gi + \frac{1}{2}C_D A \rho_a v_{ego}^2}{i_g i_0 \eta_T} r \tag{28}$$

For the desired engine torque $T_{tq,des}$, it can be directly controlled using the torque interface opened by EMS to ESC. Good control performance can be achieved by using a PID controller. It is not the research focus of this paper and will not be described further.

When the desired longitudinal acceleration is negative, the engine output torque is zero. Combining (5) and (6), the desired WCP $P_{w,des}$ can be calculated as follows:

$$P_{w,des} = \frac{-\delta m a_{des} - Gf - Gi - \frac{1}{2}C_D A \rho_a v_{ego}^2}{K_P} r \tag{29}$$

When the actual deceleration is less than the expected deceleration, the ESC is in the pressurization process, and the duty of plunger pump motor needs to be controlled. Select the system state variable $x = P_w$, the control input $u_p = U_a$, and the Euler method is used to discretize the continuous-time system defined by (10), the discrete time state-space equation can be obtained as follows:

$$x(k+1) = f_p(x(k), u_p(k)) \tag{30}$$

When the actual deceleration is greater than the expected deceleration, the ESC is in the decompression process, and the duty of the IV needs to be controlled. Select the system state variable $x = P_w$, the control input $u_v = U_c$, and the Euler method, which is used to discretize the continuous-time system defined by (22), and the discrete-time state-space equation can be obtained as follows:

$$x(k+1) = f_v(x(k), u_v(k)) \tag{31}$$

The multiple objectives in the deceleration control process include accuracy, rapidity, stability, etc., and the multi-objective optimization function $J_{DEC}$ at time $k$ is designed as follows:

$$J_{DEC}(x(k), U_k) = \sum_{i=1}^{N_p} \|x(k+i) - r(k+i)\|_L^2 + \sum_{i=1}^{N_c-1} \|u(k+i) - u_r(k+i)\|_M^2 \tag{32}$$

where $r$ and $u_r$ are reference values of system states and input and $L$ and $M$ are weight coefficient matrices.

Considering the limitations of the power supply of the vehicle chassis system and equipment, such as wheel cylinders, in summary, the optimization problem of the lower-layer deceleration controller of the ACC system under the constraints can be described as follows:

$$\begin{aligned}
&\min_{U(k)} J_{DEC}(x(k), U_k) \\
&s.t. \ U_{\min} \le U \le U_{\max} \\
&\quad\quad \Delta U_{\min} \le \Delta U \le \Delta U_{\max} \\
&\quad\quad x_{\min} \le x \le x_{\max}
\end{aligned} \tag{33}$$

where $U_{min}$ and $U_{max}$ and $\Delta U_{min}$ and $\Delta U_{max}$ are the upper and lower limits of the vehicle supply voltage and its change rate, respectively. $x_{min}$ and $x_{max}$ are the upper and lower limits of WCP.

## 4. Simulation and Experiment

In order to verify the performance of the proposed ACC algorithm, a joint simulation platform of Carsim-MATLAB/Simulink was built, and a variety of test scenarios were selected for simulation. The simulation parameters are chosen as follows: the sampling interval is selected to be 0.1 s, the prediction horizon length is selected to be 3 s, the control horizon length is selected to be 0.3 s, the weight matrices are Q = diag(0.75, 1), and R = 1. On the one hand, the impact of vehicle parameter changes on the control performance is compared with conventional MPC controllers. On the other hand, the ACC performance is compared with the conventional PID control algorithm. To compare the difference in

performance between the two controllers, the NMPC lower-layer deceleration controller was used in both simulations.

### 4.1. Simulation Results

### 4.1.1. Testing Results of Scenario A

The initial velocity of the front vehicle is 25 m/s, and the acceleration of the front vehicle changes sinusoidally: the acceleration amplitude is 0.5 m/s$^2$ and the angular frequency is 0.2 rad/s; the initial velocity of the ego vehicle is 20 m/s; and the initial relative distance between the two vehicles is 40 m.

Mass change is a common internal disturbance for vehicle parameters. In order to verify the control performance under typical vehicle parameter changes, simulations are carried out with different sprung masses in test scenario A. The suffixes A, B, and C represent the three tests, respectively. Among them, test A is the reference working condition using rated mass; test B and test C are MPC using feedback correction and not using feedback correction, respectively; and their masses are 1.5 times the rated mass. Figure 7 shows the three test results in this scenario, respectively. Through the comparison, it can be seen intuitively that the speed tracking performance and distance tracking performance in test A are the best, while the performance in test B is better than test C under the influence of vehicle mass changes. After reaching the cycle following stage at 20 s, the statistics show that the speed error amplitudes in tests A, B, and C are 0.83 m/s, 0.95 m/s, and 1.19 m/s, and the distance errors are 0.52 m, 0.89 m, and 1.75 m, respectively. This also shows that MPC based on feedback correction has better robustness.

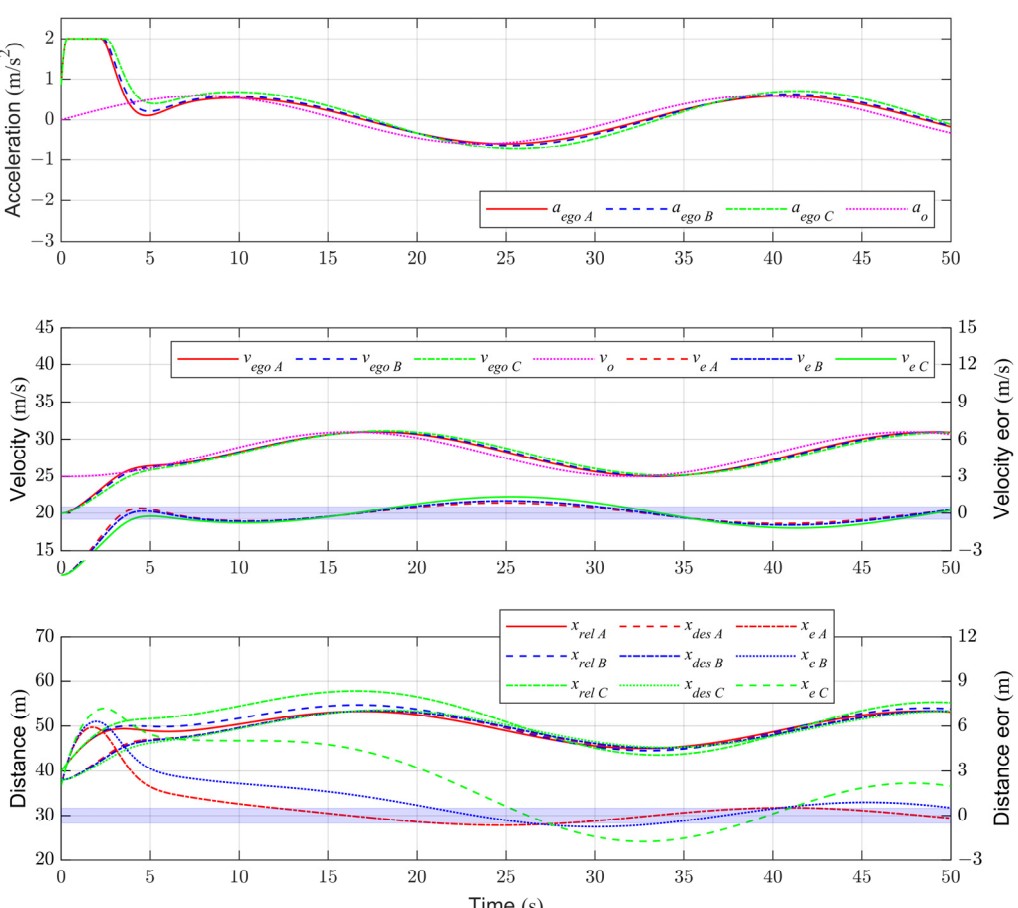

**Figure 7.** Testing results of scenario A with sprung masse change.

Figure 8 shows the motion state of the two vehicles under the action of the two controllers, respectively, in this scenario. In the acceleration stage of 0–5 s, the acceleration

and braking behavior of the ego vehicle under the PID controller is more severe in the initial stage, with a maximum speed error of 1.51 m/s and oscillation. Based on the MPC controller, the vehicle speed error quickly converges, with a maximum speed error of 0.47 m/s, which is 31% of the PID controller. There is a similar phenomenon for the relative distance error. Under the action of the PID controller, there is a maximum overshoot of ca. 3 m for relative distance error at 6.9 s. After reaching the cycle following stage at 23 s, the speed error amplitude under the action of the MPC controller is calculated to be 0.56 m/s, which is 81% of that under the action of the PID controller, and it is 0.69 m/s. The relative distance error amplitude under the action of MPC controller is 0.43 m, which is 64% of that under the action of PID controller, and it is 0.67 m.

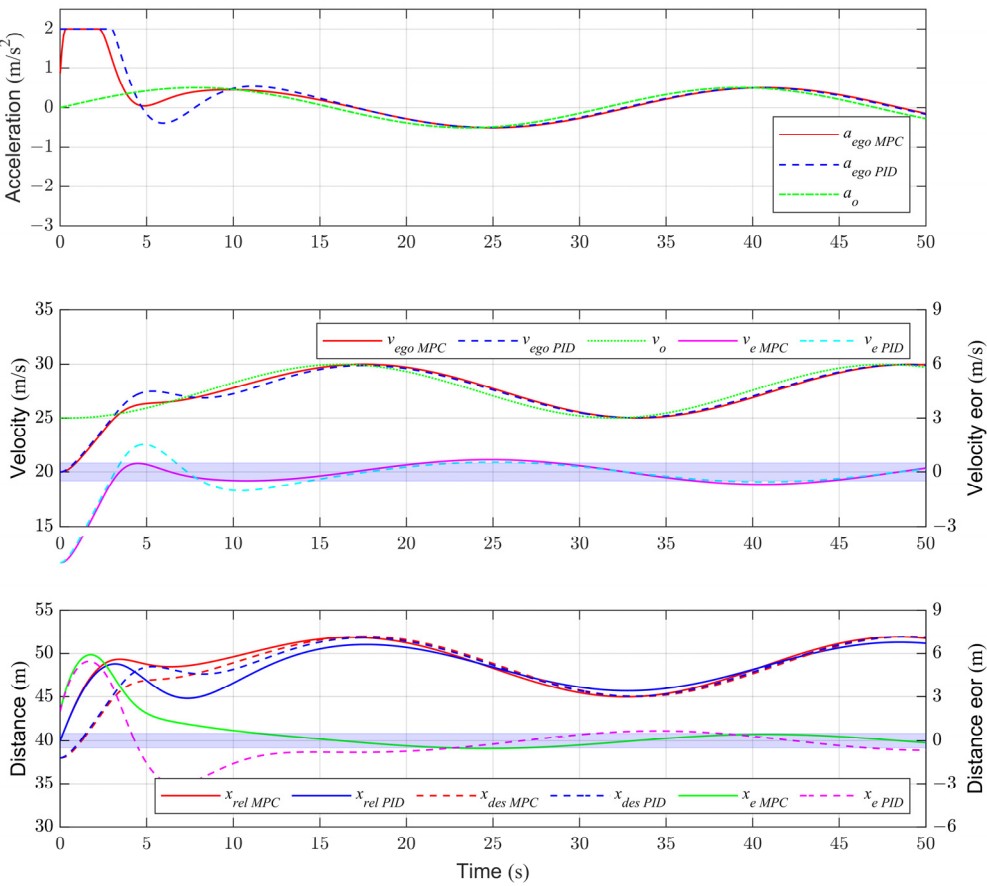

**Figure 8.** Testing results of scenario A by different controllers.

### 4.1.2. Testing Results of Scenario B

The initial speed of the front vehicle is 20 m/s, and the acceleration of the front vehicle changes stepwise several times. During the period of 10–20 s, the front vehicle accelerates to 35 m/s with an acceleration of 1.5 m/s². During the period of 30–35 s, the front vehicle decelerates to 25 m/s with a deceleration of −2 m/s². The initial speed of the ego vehicle is 20 m/s, and the initial relative distance between the two vehicles is 40 m.

Figure 9 shows the motion state of the two vehicles under the action of the two controllers in this scenario. During the constant speed stage of 0–10 s, the acceleration and braking amplitudes of the ego vehicle are greater under the PID controller, and the MPC controller is more stable. After that, when the front vehicle acceleration changes four times, the PID controller has a larger overshoot compared to the MPC controller, which will affect driving comfort. Taking the acceleration stage at 10–20 s as an example, the acceleration overshoot under the MPC controller is 0.05 m/s², which is 27.8% of that under the PID controller, which is 0.18 m/s². In addition, during the constant speed stage of 20–30 s, the MPC controller converges the speed error and relative distance error to the allowable range

at 21.9 s, while the PID controller fails to control the distance error in a timely manner after converging the speed error to the allowable range at 21.7 s. Therefore, the MPC controller can track the desired distance and desired speed faster at the same time. The same effect is achieved during the constant speed stage of 35–50 s.

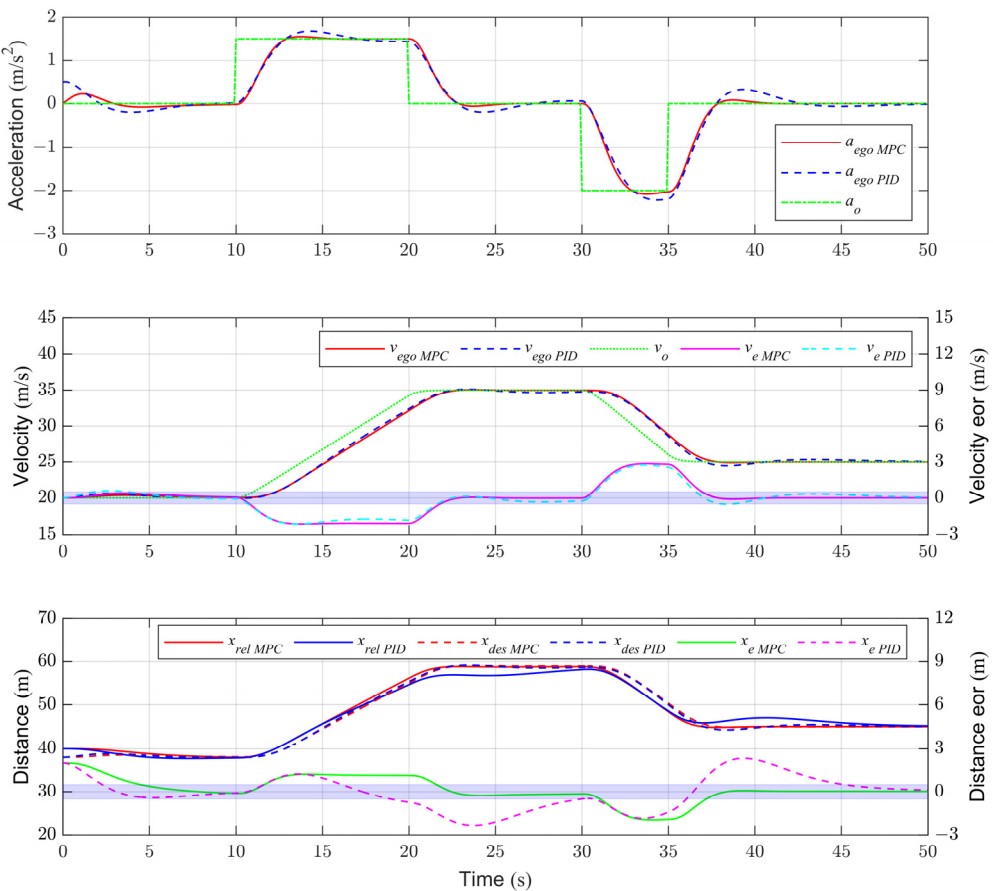

**Figure 9.** Testing results of scenario B by different controllers.

### 4.2. Experiments Results

To verify the performance of the proposed ACC control algorithm, a test vehicle was modified for real vehicle testing. As shown in Figure 10, the test vehicle and the front vehicle are in the test field. The test road surface is asphalt pavement after rain, and its road adhesion coefficient has a range of uncertainties. The test vehicle is equipped with a millimeter-wave radar and a front-facing camera to obtain the relative motion status of the front vehicle. The torque interface of the EMS system is open, and the desired torque can be obtained through the controller area network (CAN) to control the engine torque. The ego vehicle is equipped with a BBW system based on the ESC system.

The road test includes two parts, one is to test the performance difference between the NMPC controller and the PID controller in vehicle deceleration control, the other part is to test the performance difference between the proposed MPC controller and the traditional PID controller in ACC control. Some key parameters of the sensors are shown in Table 1.

**Table 1.** Key parameters of the sensors.

| Parameters | Values |
|---|---|
| Data update frequency | 15–20 Hz |
| Relative distance accuracy | ±0.4 m |
| Relative velocity accuracy | ±0.1 km/h |

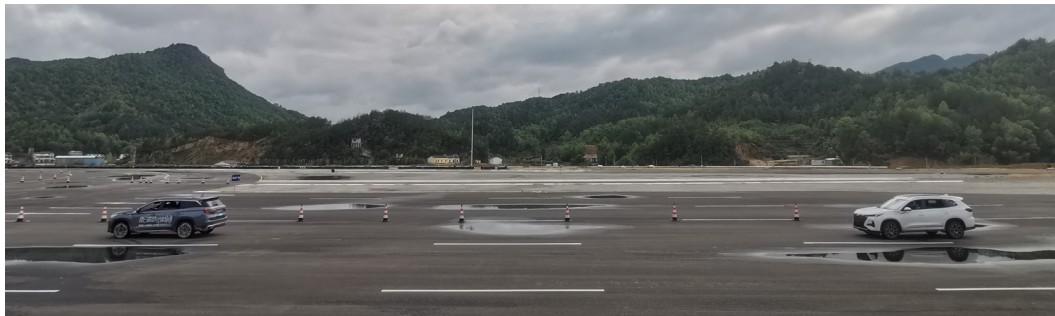

**Figure 10.** Test environment for ACC system.

### 4.2.1. Experiments results of Deceleration Controller

Figure 11 shows the step response of the NMPC deceleration controller and the PID deceleration controller. The overshoot of the NMPC deceleration controller is 0.21 m/s$^2$, which is smaller than that of the PID controller at 0.61 m/s$^2$. At the same time, the steady-state error of the deceleration under the NMPC controller approaches zero, while the deceleration under the PID controller has a certain degree of vibration. Figure 12 shows the ramp response of the NMPC deceleration controller and PID deceleration controller. The NMPC deceleration controller has a more stable deceleration change and a smaller overshoot than the PID deceleration controller.

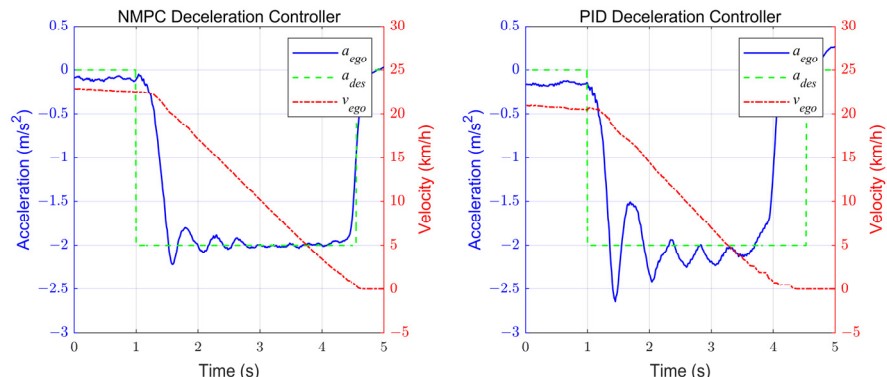

**Figure 11.** Step response of the different deceleration controllers in experiments.

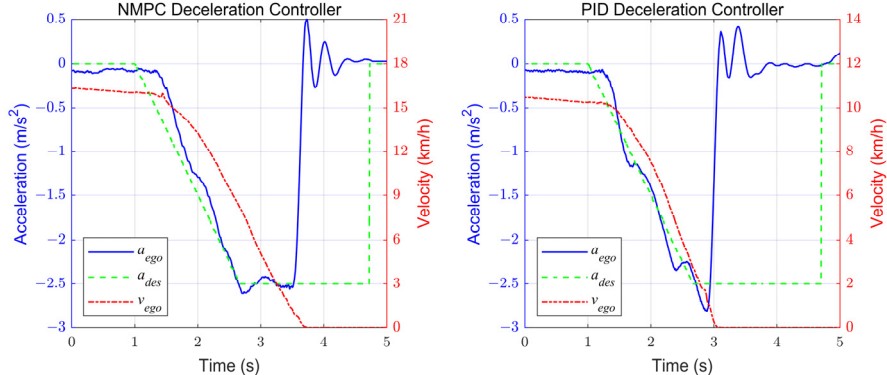

**Figure 12.** Ramp response of the different deceleration controllers in experiments.

### 4.2.2. Experiments Results of ACC Strategies

The ACC test scenario is shown in Figure 10. The initial speed of the ego vehicle and the front vehicle is 10 m/s, and the initial relative distance between the two vehicles is about 25 m. The speed of the front vehicle is shown in Figure 13, which includes an acceleration process and a deceleration process. The allowable range of velocity error is

$\pm0.5$ m/s, and the allowable range of distance error is $\pm0.5$ m. In order to ensure the consistency of the motion state of the front vehicle during multiple tests, after recording the motion data of the front vehicle during the first test, the virtual front vehicle is used in subsequent tests, and the processed relative motion state data are directly input into the ACC controller.

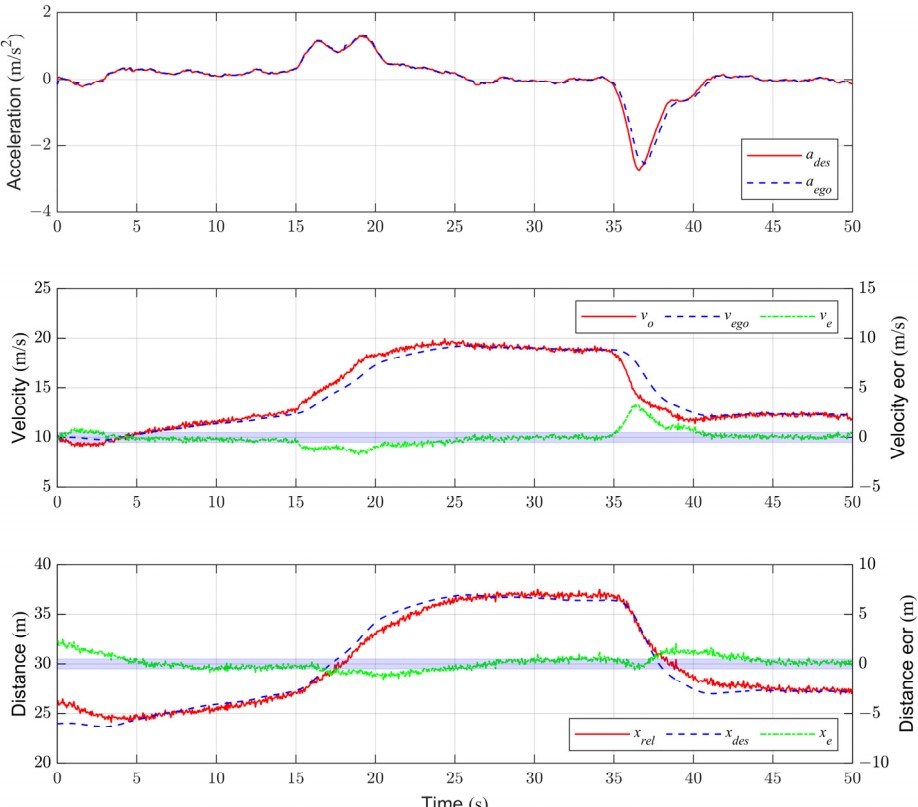

**Figure 13.** Test results of the ACC system based on the MPC controller.

Figures 13 and 14, respectively, show the performance of the ACC system based on the MPC controller and the PID controller in this test scenario. Under the PID controller, the acceleration and braking amplitude of the ego vehicle in the initial stage is larger, its initial acceleration exceeds 0.5 m/s$^2$, and the MPC controller is more stable. During the acceleration and deceleration process of the front vehicle, both the MPC controller and the PID controller can effectively track the velocity of the front vehicle after 23 s and converge the tracking error within the allowable range. However, at the same time, the MPC controller can track the target distance better and control the error within the allowable range after 26 s, while the PID controller has a large tracking error and cannot control the error to the allowable range in time. In addition, the designed NMPC deceleration controller can accurately track the desired deceleration, which provides a guarantee for the performance of the ACC system.

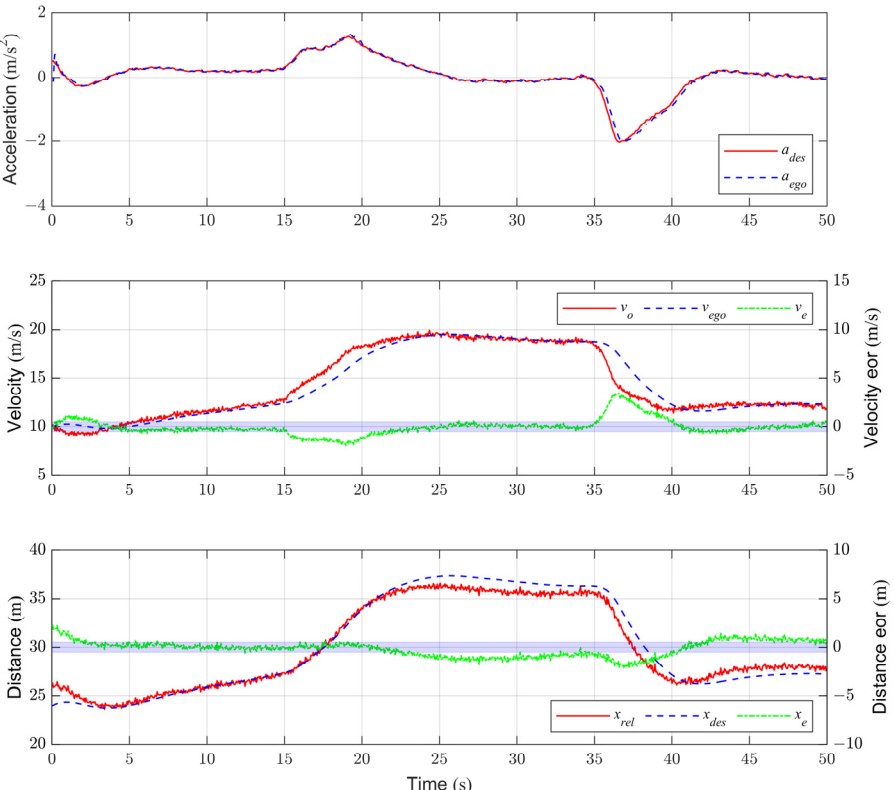

**Figure 14.** Test results of ACC system based on the PID controller.

## 5. Conclusions

This paper proposes a hierarchical ACC system framework based on the BBW system. On the one hand, the upper-layer controller is designed based on the MPC algorithm to realize the multi-objective cooperative control of the desired vehicle speed and the desired relative distance. On the other hand, by analyzing the dynamic model of the vehicle and the braking system model of the HCU, an NMPC deceleration controller is proposed to achieve a stable and fast response to the deceleration.

The simulation and experimental results show that the proposed ACC control strategy can achieve coordinated control of multiple control objectives at the same time. Additionally, compared with traditional PID controllers, it can more reliably control the vehicle's motion state when the front vehicle state suddenly changes. At the same time, compared with the traditional PID deceleration controller, the NMPC deceleration controller can also track the desired deceleration more accurately and smoothly. Therefore, the comfort and reliability of the control process can be effectively improved.

As an assisted driving system that improves comfort, the ACC system needs to consider many other optimization objectives, such as fuel economy. Future research directions will focus on improving the proposed ACC strategy, verifying the stability and robustness of the system in more test scenarios, optimizing the design parameters and control logic of the hydraulic control unit, and improving the quality of deceleration control.

**Author Contributions:** Conceptualization, H.M. and L.L.; methodology, H.M.; software, H.M.; validation, H.M., M.M. and Y.Z.; formal analysis, M.M.; investigation, Y.Z.; resources, L.L.; data curation, H.M.; writing—original draft preparation, H.M.; writing—review and editing, H.M.; visualization, M.M.; supervision, L.L.; project administration, L.L.; funding acquisition, L.L. All authors have read and agreed to the published version of the manuscript.

**Funding:** This research was funded by the National Key Research and Development Program of China, grant number 2022YFB2503102.

**Institutional Review Board Statement:** Not applicable.

**Informed Consent Statement:** Not applicable.

**Data Availability Statement:** Not applicable.

**Conflicts of Interest:** The authors declare no conflict of interest. The funders had no role in the design of the study; in the collection, analyses, or interpretation of data; in the writing of the manuscript; or in the decision to publish the results.

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
