# Peer review of "A Hierarchical Control Scheme for Adaptive Cruise Control System Based on Model Predictive Control"

_actuators, doi:10.3390/act12060249_

Round 1
Reviewer 1 Report
In this paper, a hierarchical anti-disturbance cruise control architecture based on electronic stability control (ESC) system is proposed. The upper-layer controller outputs the desired longitudinal acceleration or deceleration to the lower-layer actuators. In order to improve the accuracy of model prediction and achieve coordinated control of multiple objectives, an upper-layer model prediction cruise controller is established based on feedback control and disturbance compensation. In addition, based on the hydraulic control unit (HCU) model and the vehicle longitudinal dynamics model, a lower-layer nonlinear model predictive deceleration controller is proposed, to solve the problems of pressure fluctuations and low accuracy of small deceleration when ESC is used as the actuator for ACC system. Simulation and experiment demonstrate the effectiveness of the proposed control architecture. Thanks for writing this high-quality article, but there are several problems:
1. The quality of the pictures or figures is too poor. Such as Figure 6 is not very clear, please provide a clear one.
2. Something is wrong with the formula number, like (#) instead of (#.).
3. The weight matrix in MPC has a great influence on the MPC control effect. The weight value is not given specifically in this paper. Is the weight in this paper a definite value or a variable value?
4. The problems mentioned in the abstract and state of the art include: model prediction cruise controller accuracy and Multi-objective coordinated control of speed and distance. When ESC is used as the actuator of the ACC system, there are some problems, such as large pressure fluctuation and low control accuracy in small decelerations. Considering the comprehensive interference of vehicle parameters and the uncertainty of road conditions in the process of driving, the comprehensive interference of vehicle parameters, the uncertainty of road conditions, and other factors were not added to the experiment.
5 It is mentioned that nonlinear MPC is used to reduce the influence of the vehicle nonlinear model on the control, but the experimental results clearly cannot reflect the advantages of nonlinear MPC over MPC.
The quality of the English language is moderate.
Reviewer 2 Report
In the article under review, the authors solve the problem of improving the efficiency of the adaptive cruise control system of a passenger car. The authors proposed a new control strategy for the hierarchical adaptive cruise control system.
The studies were carried out by mathematical simulation using the Matlab, CarSim and Simulink software, as well as with experimental tests.
In the Introduction and the literature review, the prerequisites for conducting research are considered in sufficient detail, an analysis of previously published studies is carried out, and the purpose of the paper is formulated. In the main parts of the paper, a mathematical description of the adaptive cruise control system is presented, including the mathematical description of vehicle dynamics and the mathematical description of the brake-by-wire system. A controller for an adaptive cruise control system based on a predictive control model is proposed. The results of the simulation and experimental tests are presented, which confirmed the better efficiency of the proposed control strategy in comparison with the traditional PID controller.
However, during the review, I noticed the low quality of most of the figures presented in the paper. This should be corrected.
In addition, the authors should have more clearly formulated the scientific novelty of the presented developments. Otherwise, in its present form, the paper looks like the application of well-known methods to a known object.
In general, the presented paper will be of interest to scientists and researchers, specialists in the field of vehicle control. I would like to thank the authors for the quality research and congratulate the authors on a well-prepared article. I recommend this article for publication after minor revision.
Round 2
Reviewer 1 Report
After the authors's modification, I agree to accept this article.